# An Approximation Algorithm for Graph Label Selection

**Josia John** [1]  **Simon Meierhans** [1]  **Maximilian Probst Gutenberg** [1]

## Abstract

In the graph label selection problem, one is given an $n$-vertex graph and a budget $k$, and seeks to select $k$ vertices whose labels enable accurate prediction of the labels on the remaining vertices. This problem formalizes distilling a small representative set from the whole graph.

We present the first $\tilde{O}(\log^{1.5} n)$-approximation algorithm for graph label selection under the standard budget constraint. Prior work either relies on resource augmentation, allowing substantially more than $k$ labeled vertices, or consists primarily of heuristics without provable guarantees.

Finally, we demonstrate that practical heuristic variants of our algorithm scale to significantly larger graphs than previous methods, while essentially retaining their quality.

## 1. Introduction

Selecting a representative small subset of a large data set is a crucial task in various areas of machine learning, such as speeding up training (Nguyen et al., 2021; Yang et al., 2024) and selecting diverse additional context for passing to a large language model (LLM) for improved inference (Liskavets et al., 2025; Cheng et al., 2024; Bateni et al., 2025).

In active learning, the training set, i.e. the *labeled set*, is not fixed but rather chosen by the learning algorithm itself to be as representative of the entire dataset as possible. For pairwise similarities between data points represented as a graph $G = (V, E, w)$, Guillory & Bilmes (2009) introduced a natural objective for selecting such a labeled set. They fix $k$ to be the maximum number of data points to be labeled, and aim to rule out that a large cluster of unlabeled points with few connections to the rest of the data set ex-

ists. Formally, they focus on selecting a set of nodes $L$, under the cardinality constraint $|L| \leq k$, which maximize the objective

$$\Psi(L) := \min_{C \subseteq V \setminus L} \frac{w(C, V \setminus C)}{|C|}.$$

This yields the Graph Label Selection Problem (GLS), which requires finding

$$\text{OPT}_k := \max_{L \subset V : |L| \leq k} \Psi(L).$$

Despite its early and natural formulation, algorithms with theoretical guarantees were lacking for a long time. Guillory & Bilmes (2009) give some practical heuristics to maximize $\Psi(L)$. Cesa-Bianchi et al. (2010) presented an algorithm with theoretical guarantees on unweighted trees.

Recently, Cohen-Addad et al. (2025) showed that the problem is NP-hard. Therefore, it is natural to consider approximation algorithms for this problem. There are two natural notions of approximation for this problem. The weaker notion, an $\alpha$-resource augment algorithm, uses budget $\alpha \cdot k$ but is competitive with $\text{OPT}_k$, i.e., the optimal value obtained with much smaller budget $k$. An $O(\log n)$-resource algorithm was recently given in (Cohen-Addad et al., 2025).

In (Cohen-Addad et al., 2025), it is stated as an interesting open problem to achieve a stronger and more natural version of approximation: they ask whether an algorithm exists that uses the given $k$ and is $\beta$-competitive in $\text{OPT}_k$, i.e. the solution has quality at least $\text{OPT}_k/\beta$. In this article we resolve this question by presenting an efficient $\tilde{O}(\log^{1.5} n)$ approximation algorithm for maximizing $\Psi(L)$, which does not rely on resource augmentation. Additionally, a relaxed version of our new algorithm has better runtime and can thus be scaled to larger instances.

**Theorem 1.1.** *Given a graph $G = (V, E, w)$ with weights in $[1, poly(n)]$, and a budget $k \in \mathbb{N}$, there exists a polynomial-time algorithm that returns a set $L'$ with*

- $|L'| \leq k$ *and*

- $\beta \cdot \Psi(L') \geq OPT_k$

*where $\beta \in \tilde{O}(\log^{1.5} n)$.*

---

[1]Department of Computer Science, ETH Zurich, Zurich, Switzerland. Correspondence to: Josia John <jojohn@ethz.ch>, Simon Meierhans <simon.meierhans@inf.ethz.ch>, Maximilian Probst Gutenberg <maximilian.probst@inf.ethz.ch>.

*Proceedings of the 43$^{rd}$ International Conference on Machine Learning*, Seoul, South Korea. PMLR 306, 2026. Copyright 2026 by the author(s).

To achieve our results, we go beyond previous greedy approaches that select labeled points one at a time. Instead, our algorithm captures global interactions between the labeled points, which is necessary for obtaining a guarantee on the approximation with a fixed budget.[1]

Additionally, our algorithm is optimal on weighted trees (See Section A) and seamlessly extends to vertex importances (See Section B).

We complement this theoretical result with a proof-of-concept experiment. We give an implementation of the algorithm where we replace the key primitive of finding sparse cuts with simple heuristics. Even with these simple heuristics, our algorithm essentially matches the quality of previous algorithms (Cohen-Addad et al., 2025; Guillory & Bilmes, 2009; Cesa-Bianchi et al., 2010) while being significantly more scalable.

**Overview.** Previous algorithms build the set $L$ one by one. But the objective $\Psi(L)$ is neither submodular nor supermodular (Cohen-Addad et al., 2025). This means it is unclear how to choose the next vertex without resorting to heuristics. We take a new approach and select the set $L$ using Dynamic Programming.

To be able to apply Dynamic Programming, we reduce the graph label selection problem twice. First, we reduce it to solving a problem on a weighted binary tree using a *tree cut sparsifier*. A tree cut sparsifier is a tree $T$ spanning the same vertex set as the input graph such that **every** cut $(S, V \setminus S)$ has approximately the same weight in $G$ and in $T$. In (Räcke & Shah, 2014), a polynomial-time algorithm is given that constructs a tree cut sparsifier with approximation factor $\beta = \tilde{O}(\log^{1.5} n)$. We give the reduction in Section 3. In Section 4, we use a similar reduction as in (Cohen-Addad et al., 2025) to reduce to a flow problem. However, while Cohen-Addad et al. (2025) need to solve this flow problem on a general graph, for us, it suffices to solve it on a binary tree. This allows us, in the final step in Section 5, to solve the flow problem via Dynamic Programming (bypassing the need to solve the maximum flow problem).

**Related Work.**

- Active learning: We restrict our discussion of related work to graph-based approaches, and refer the reader to (Settles, 2009; Ren et al., 2021) for surveys on classical and deep active learning, respectively.

  The problem of selecting $L$ such that it maximizes our knowledge about the other vertices in the graph has been studied thoroughly (Zhu et al., 2003; Guillory & Bilmes, 2009; Cesa-Bianchi et al., 2010; Dasarathy et al., 2015) from both combinatorial and deep learning (Mac Aodha et al., 2014; Kushnir & Venturi, 2020; Hu et al., 2020; Zhang et al., 2022b;a) based angles.

- Tree Cut Sparsifiers: The first polynomial-time algorithms for tree cut sparsifiers with polylogarithmic approximation was given by Harrelson et al. (2003). The above result by Räcke & Shah (2014) improves the approximation factor significantly. A lower bound of $\Omega(\log n)$ in the approximation, even existentially, is sketched in (Räcke & Shah, 2014). More recently, near-linear time algorithm achieving an approximation of $\tilde{O}(\log^2 n)$ were given (Räcke et al., 2014; Agassy et al., 2025; Henzinger et al., 2025).

- Diversity Sampling: Our algorithm can be seen as a form of diversity sampling. We refer the reader to (Anand et al., 2025) for a recent graph based algorithm in this space.

## 2. Preliminaries

**Misc.** In this article, $\tilde{O}(f(n))$-notation suppresses logarithmic factors in $f(n)$, i.e. is in $O(f(n) \log^c(f(n)))$ for some constant $c > 0$.

**Graphs.** We consider undirected, weighted graphs $G = (V, E, w)$ where $V$ denotes the vertex set and $E \subseteq V \times V$ denotes the edge set. The function $w : E \mapsto \mathbb{R}$ contains the non-negative edge weights. We let $n$ denote the number of vertices $|V|$.

For some set $A \subseteq V$, we denote by $G[A]$ the subgraph induced by $A$, i.e. the graph obtained when restricting to the set of vertices in $A$.

**Trees.** We use $\mathcal{L}_T$ to denote the leaves of a tree $T$. If the tree is rooted, we denote the subtree of vertex $v$ by $T_v$.

**Cuts.** We define the weight of a cut $A \subseteq V$ in some graph $G = (V, E, w)$:

$$w_G(A, V \setminus A) \coloneqq \sum_{\substack{(u,v) \in E \\ u \in A, v \in V \setminus A}} w(u, v).$$

We also define the minimum cut separating two disjoint sets $A, B \subseteq V, A \cap B = \emptyset$:

$$\lambda_G(A, B) \coloneqq \min_{A \subseteq S \subseteq V \setminus B} w(S, V \setminus S).$$

**Graph Label Selection (GLS)** For convenience, we restate the definition of the Graph Label Selection Problem

---

[1]Star graphs are a simple example that highlight this behavior. If the budget is $k = n - 1$, it is imperative not to select the center of the star which is the obvious greedy choice.

here. We define the optimal solution to GLS on a graph $G = (V, E, w)$ as

$$\text{OPT}_k := \max_{L \subset V : |L| \leq k} \Psi(L),$$

where

$$\Psi(L) := \min_{C \subseteq V \setminus L} \frac{w(C, V \setminus C)}{|C|}.$$

## 3. Reducing to Binary Tree

Our algorithm reduces the graph label selection problem from a general graph to a related problem on a binary tree. We do so by first constructing a tree cut sparsifier for the given graph.

**Definition 3.1** (Tree Cut Sparsifier)**.** A tree $T = (V_T, E_T, w_T)$ is an $\alpha$ tree cut sparsifier for the graph $G = (V, E, w)$ if the leaves $\mathcal{L}_T = V$, and for all $A \subseteq V$:

$$w(A, V \setminus A) \leq \lambda_T(A, \mathcal{L}_T \setminus A) \leq \alpha \cdot w(A, V \setminus A).$$

We now translate the graph label selection problem to the tree cut sparsifier. We call this new problem the leaf label selection problem. For this we need the notion of sparsity on the leaf set.

**Definition 3.2** (Sparsity on Subset)**.** Given a graph $G = (V, E, w)$, we define the sparsity of $A$ in a superset $C \supset A$:

$$\psi_G^C(A) := \frac{\lambda_G(A, C \setminus A)}{|A|}.$$

**Definition 3.3** (Leaf Label Selection Problem (LLS))**.** Given a tree $T = (V, E, w)$ and a budget $k \in \mathbb{N}$, find a set $L \subseteq \mathcal{L}_T, |L| \leq k$ that maximizes the objective

$$\widehat{\Psi}_T(L) := \min_{S \subseteq \mathcal{L}_T \setminus L} \psi_T^{\mathcal{L}_T}(S).$$

Note that this is different from the graph label selection problem in that we must choose $L$ as a subset of the leaf set $\mathcal{L}_T$.

Now we reduce the GLS problem on general graphs to an LLS problem on trees with loss of factor $\alpha = \tilde{O}(\log^{1.5} n)$ in the approximation.

**Lemma 3.4.** *Given a graph $G = (V_G, E_G, w_G)$ and a corresponding $\alpha$ tree cut sparsifier $T = (V_T, E_T, w_T)$. For any solution $L \subseteq \mathcal{L}_T = V$, we have:*

$$\Psi_G(L) \leq \widehat{\Psi}_T(L) \leq \alpha \cdot \Psi_G(L).$$

*Proof.* For the upper bound, we derive

$$\begin{aligned}
\widehat{\Psi}_T(L) &= \min_{S \subseteq \mathcal{L}_T \setminus L} \psi_T^{\mathcal{L}_T}(S) \\
&= \min_{S \subseteq \mathcal{L}_T \setminus L} \frac{\lambda_T(S, \mathcal{L}_T \setminus S)}{|S|} \\
&\overset{3.1}{\leq} \min_{S \subseteq V \setminus L} \frac{\alpha \cdot w_G(S, V \setminus S)}{|S|} \\
&= \alpha \cdot \min_{S \subseteq V \setminus L} \psi_G(S) = \alpha \cdot \Psi_G(L).
\end{aligned}$$

The lower bound follows analogously. $\qquad \square$

**Corollary 3.5.** *Given a graph $G = (V, E, w)$ and a corresponding $\alpha$ tree cut sparsifier $T = (V_T, E_T, w_T)$. Let $L'$ be an optimal solution to LLS on $T$. Then $L'$ is a solution for GLS with approximation factor $\alpha$.*

*Proof.* Let $L^*$ denote an optimal solution of GLS on $G$. Using Lemma 3.4, and optimality of $L'$ for LLS, we obtain

$$\alpha\Psi_G(L') \geq \widehat{\Psi}_T(L') \tag{3.4}$$

$$\widehat{\Psi}_T(L') \geq \widehat{\Psi}_T(L^*) \geq \Psi_G(L^*) \quad \text{(optimality, 3.4)}$$

$$\implies \alpha\Psi_G(L') \geq \Psi_G(L^*).$$

$$\square$$

We have reduced GLS to LLS on trees. Now, we show that solving LLS on a binary tree is sufficient by constructing a binary tree cut sparsifier via standard reductions (Räcke & Shah, 2014).

**Lemma 3.6.** *Given an $\alpha$ tree cut sparsifier $T$ of some graph $G$, we can compute a binary tree $T'$ which is an $\alpha$ tree cut sparsifier of $G$ too. This can be done in linear time.*

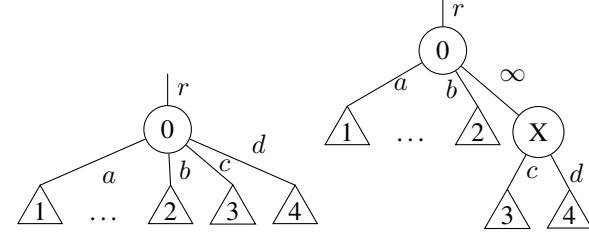

*Figure 1.* Decreasing degree by 1 by introducing a new vertex $X$ with $\infty$ edge weight.

*Proof.* Given an internal vertex $v$ with degree larger than 3, we can add an auxiliary vertex $x$ as a child of $v$ with infinite edge weight and attaching two of $v$'s children to $x$ instead (see Figure 1). This introduces no new vertices with degree larger than 3 and reduces the degree of $v$ by one, therefore iteratively applying this procedure results in a binary tree. Finally, we show that the resulting tree remains a tree cut sparsifier. We do so by showing it for a single step and applying induction. Firstly, any cut in the original

construction can be reproduced in the new construction by cutting the same edges. Secondly, any cut with finite cut size in the new construction does not use the $\infty$ edge. Therefore, the requirement

$$w(A, V \setminus A) \le \lambda_T(A, \mathcal{L}_T \setminus A) \le \alpha \cdot w(A, V \setminus A)$$

still holds. $\qquad\square$

Combining the above we obtain the following lemma.

**Lemma 3.7** (Reduction of GLS to LLS). *Given an $\alpha$ tree cut sparsifier over $O(n)$-vertices of $G$, then GLS on $G$ with an approximation factor of $\alpha$ can be reduced to solving LLS on an $O(n)$-vertices binary tree instance after processing in $O(n)$ time.*

*Proof.* This follows immediately from Corollary 3.5 and Lemma 3.6. $\qquad\square$

In the rest of this article, we therefore focus on solving LLS on binary trees.

## 4. Reducing to Flow Problem

In (Cohen-Addad et al., 2025), they construct a flow gadget inspired by the densest subgraph problem to obtain a resource augmented algorithm for graph label selection. We observe that a similar flow problem can be solved exactly on the binary tree $T$ via dynamic programming. For this purpose, we construct a graph $T_{L,\tau}$ such that for any parameter $\tau \ge 0$, $\widehat{\Psi}_T(L) \ge \tau$ if and only if the $s$-$t$-maxflow problem on $T_{L,\tau}$ has value $n \cdot \tau$ where $n = |\mathcal{L}_T|$ denotes the number of leaves in $T$.

**Definition 4.1** (Flow Graph). Given a tree $T = (V, E, w)$, a set $L \subseteq \mathcal{L}_T$, and a threshold $\tau \in \mathbb{R}$. We construct $T_{L,\tau}$:

- Vertex Set: $V \cup \{s, t\}$

- Graph Copy: Every edge $e \in E$ is also present in $T_{L,\tau}$ with the same weight.

- Source edges: For every leaf $v \in \mathcal{L}_T$, there is an edge $(s, v)$ with weight $\tau$ in $T_{L,\tau}$.

- Sink edges: For every chosen leaf $v \in L$, there is an edge $(v, t)$ with weight $\infty$ in $T_{L,\tau}$.

We start by showing that if the $s$-$t$-mincut is less than $\tau \cdot n$, then $L$ gives a solution with value less than $\tau$.

**Lemma 4.2.** *If $\lambda_{T_{L,\tau}}(\{s\}, \{t\}) < \tau \cdot n$, then $\widehat{\Psi}_T(L) < \tau$.*

*Proof.* By the definition of mincut, there is some $S'$ such that:

$$\lambda_{T_{L,\tau}}(\{s\}, \{t\}) = w_{T_{L,\tau}}(S' \cup \{s\}, (V \setminus S') \cup \{t\})$$

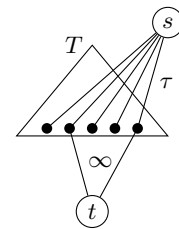

*Figure 2.* The structure of $T_{L,\tau}$.

We notice that $S', L$ must be disjoint because all vertices in $L$ have an edge of infinite capacity to $t$. We can write the size of this cut as the sum of two parts, the cut in the original tree $T$ plus the edges from the new sink $s$ to vertices in $(V \setminus S') \cup \{t\}$. This yields

$$
\begin{aligned}
w_{T_{L,\tau}}&(S' \cup \{s\}, (V \setminus S') \cup \{t\}) \\
&= w_T(S', V \setminus S') + w_{T_{L,\tau}}(\{s\}, (V \setminus S') \cup \{t\}).
\end{aligned}
$$

The second part of the sum consists of the edges from $\{s\}$ to the leaves $\mathcal{L}_T \setminus S'$. Each such edge has weight $\tau$ by construction, so the sum can be written as

$$
\begin{aligned}
w_{T_{L,\tau}}&(S' \cup \{s\}, (V \setminus S') \cup \{t\}) \\
&= w_T(S', V \setminus S') + \tau \cdot |\mathcal{L}_T \setminus S'|.
\end{aligned}
$$

Now we have

$$\lambda_{T_{L,\tau}}(\{s\}, \{t\}) = w_T(S', V \setminus S') + \tau \cdot |\mathcal{L}_T \setminus S'|.$$

Plugging this into the LHS of the lemma statement $\lambda_{T_{L,\tau}}(\{s\}, \{t\}) < \tau \cdot n$, we get

$$
\begin{aligned}
w_T(S', V &\setminus S') + \tau \cdot |\mathcal{L}_T \setminus S'| < \tau \cdot n \\
&\implies \quad w_T(S', V \setminus S') < \tau \cdot (n - |\mathcal{L}_T \setminus S'|).
\end{aligned}
$$

Then, let $S = S' \cap \mathcal{L}_T$. We have

$$\lambda_T(S, \mathcal{L}_T \setminus S) \le w_T(S', V \setminus S') < \tau \cdot (n - |\mathcal{L}_T \setminus S'|) = \tau \cdot |S|.$$

So we have $\psi_T^{\mathcal{L}_T}(S) < \tau$. Because $S, L$ are disjoint, this gives $\widehat{\Psi}_T(L) < \tau$ and concludes the proof. $\qquad\square$

And now we show the other direction. If the solution $\widehat{\Psi}_T(L) < \tau$, then the $s$-$t$-maxflow is less than $\tau \cdot n$.

**Lemma 4.3.** *If $\widehat{\Psi}_T(L) < \tau$, then $\lambda_{T_{L,\tau}}(\{s\}, \{t\}) < \tau \cdot n$.*

*Proof.* By the definition of $\widehat{\Psi}$, there is some $S$ disjoint from $L$ such that:

$$
\begin{aligned}
\widehat{\Psi}_T(L) &= \psi_T^{\mathcal{L}_T}(S) < \tau \\
\implies \quad \lambda_T&(S, \mathcal{L}_T \setminus S) < \tau \cdot |S|.
\end{aligned}
$$

Now, by the definition of a minimum cut, there is some $S'$ with $S \subseteq S' \subseteq V \setminus (\mathcal{L}_T \setminus S)$ such that:

$$w_T(S', V \setminus S') < \tau \cdot |S|.$$

We analyze $w_{T_{L,\tau}}(S' \cup \{s\}, (V \setminus S') \cup \{t\})$. As above, we can split this up:

$$w_{T_{L,\tau}}(S' \cup \{s\}, (V \setminus S') \cup \{t\})$$
$$= w_T(S', V \setminus S') + w_{T_{L,\tau}}(\{s\}, (V \setminus S') \cup \{t\}).$$

We now bound $w_{T_{L,\tau}}(\{s\}, (V \setminus S') \cup \{t\})$ as in the proof of Lemma 4.2. We obtain

$$w_{T_{L,\tau}}(S' \cup \{s\}, (V \setminus S') \cup \{t\})$$
$$= w_T(S', V \setminus S') + \tau \cdot |\mathcal{L}_T \setminus S'|$$
$$< \tau \cdot |S| + \tau \cdot |\mathcal{L}_T \setminus S'|.$$

And because $S \subseteq S'$, we have $\tau \cdot |S| + \tau \cdot |\mathcal{L}_T \setminus S'| \leq \tau \cdot |\mathcal{L}_T| = \tau \cdot n$. This directly implies $\lambda_{T_{L,\tau}}(\{s\}, \{t\}) < \tau \cdot n$ and concludes the proof. □

We finally combine the two lemmas to get an equivalence statement.

**Corollary 4.4.**

$$\widehat{\Psi}_T(L) \geq \tau \iff \lambda_{T_{L,\tau}}(\{s\}, \{t\}) = n \cdot \tau$$

*Proof.* We have $\lambda_{T_{L,\tau}}(\{s\}, \{t\}) \leq n \cdot \tau$ because the degree of $s$ is $n \cdot \tau$. Therefore, the corollary follows from Lemma 4.2 and Lemma 4.3. □

By the mincut-maxflow duality, we can obtain the value of the minimum cut by solving the $s$-$t$-maxflow problem on $T_{L,\tau}$.

This motivates us to define a new problem which aims to construct a set $L$ such that $T_{L,\tau}$ has $s$-$t$-maxflow $\tau \cdot n$. We call this new problem the "sink selection problem".

**Definition 4.5** (Sink Selection Problem). Given a tree $T = (V, E, w)$ and a parameter $\tau$, find $L \subseteq \mathcal{L}_T$ with minimal $|L|$, such that the $s$-$t$-maxflow in $T_{L,\tau}$ has value $n \cdot \tau$.

We observe that the sink selection problem is monotone in $|L|$.

**Lemma 4.6.** *For any tree $T$, and two parameters $\tau_1 \leq \tau_2$, let $L_1, L_2$ be the optimal solutions to the sink selection problem on $T$ with parameters $\tau_1$ and $\tau_2$ respectively. Then $|L_1| \leq |L_2|$*

*Proof.* We show that if $\tau_1 \leq \tau_2$, then $L_2$ is a solution to the sink selection problem on $(T, \tau_1)$ too. By definition there is a $s$-$t$ flow in $T_{L_2,\tau_2}$ with value $n \cdot \tau_2$. Now we multiply the flow on every edge with $\tau_1/\tau_2$. This results in a $s$-$t$ flow

of value $n \cdot \tau_1$. In $T_{L_2,\tau_2}$ all edges outgoing from $s$ were saturated, so now they have flow $\tau_1$. We can change the capacity of these edges to $\tau_1$, giving the graph $T_{L_2,\tau_1}$ with flow value $n \cdot \tau_1$. □

This monotonicity allows us to search for the optimal $\tau$ using binary search.

**Lemma 4.7.** *Given an algorithm that solves the sink selection problem, we can construct an algorithm solving the leaf label selection problem.*

*Proof.* We binary search for $\tau$. In every iteration we check whether the solution $L$ to the sink selection problem on $(T, \tau)$ has more or less than $k$ vertices. We can use binary search because we have shown that $|L|$ is monotone in $\tau$ in Lemma 4.6.

Let us assume that the weights of the graph $T$ are integral and lie between 1 and $W$. We call $W$ the aspect ratio of the edge weights. $T$ is connected, so we have $\frac{w(C, V \setminus C)}{|C|} \geq \frac{1}{|C|} \geq \frac{1}{n}$, for every $C \subseteq V \setminus L$. So $\widehat{\Psi}(L) \geq 1/n$. Further, for every $C \subseteq V \setminus L$, we have $\frac{w(C, V \setminus C)}{|C|} \leq \frac{n^2 \cdot W}{|C|} \leq n^2 \cdot W$, so $\widehat{\Psi}(L) \leq n^2 \cdot W$. This means that the binary search runs in $O(\log(n^2 \cdot W)) = O(\log(n \cdot W))$ time and finds the optimal solution in the polynomially bounded search space.[2] □

In the next section we show how we can solve the sink selection problem using dynamic programming and thus resolve the final missing piece.

**Remark on prior work.** After completing and submitting this work, we became aware that a problem equivalent to sink selection was previously solved by Andreev et al. (2009). We give our proof for completeness.

## 5. Solving the Flow Problem using Dynamic Programming

In this section we provide an algorithm and proof for the sink selection problem. To describe our algorithm, we need the tree $T$ to be rooted. Therefore, we fix a root arbitrarily on a non-leaf vertex.[3]

We first provide intuition about this problem. Given a tree $T$, and a parameter $\tau$, we interpret all leaves as sources of value $\tau$. Now we need to choose some leaves $L \subseteq \mathcal{L}_T$ as sinks of arbitrarily large capacity, such that the flow becomes routable. Of course we additionally want to choose $L$ such that $|L|$ is as small as possible.

---

[2]For a standard encoding of edge weights this could give $O(\text{poly}(n))$ runtime. However, the aspect ratio is typically not exponential.

[3]Often hierarchical decompositions such as tree cut sparsifiers give rise to a natural root.

We suggest using dynamic programming with the following state to resolve this problem. For some subtree of vertex $v$, denoted as $T_v$, and some number of allowed sinks $k$, record the maximum amount of flow we can inject into the subtree at the vertex $v$ such that it can still be routed away.

To formalize this, we extend Definition 4.1 of our flow gadget. The changes are highlighted in red:

**Definition 5.1** (Flow Graph). Given a tree $T = (V, E, w)$, and a set $L \subseteq \mathcal{L}_T$, and a threshold $\tau \in \mathbb{R}$, and a injected flow $f \in \mathbb{R}$. We construct $T_{L,\tau,f}$:

- Vertex Set: $V \cup \{s, t\}$

- Copy: Every edge $e \in E$ is also present in $T_{L,\tau,f}$ with the same weight.

- Source: For every leaf $v \in \mathcal{L}_T$, there is an edge $(s, v)$ with weight $\tau$ in $T_{L,\tau,f}$. Additionally, there is a directed edge $(s, \text{root}(T))$ with weight $f$. If $f$ is negative, the edge is directed the other way with weight $-f$.

- Sink: For every chosen leaf $v \in L$, there is an edge $(v, t)$ with weight $\infty$ in $T_{L,\tau,f}$.

Now we can formalize the dynamic programming state. For some subtree of vertex $v$, denoted as $T_v$, and some number of allowed sinks $k$, what is the most amount of flow we can inject into the subtree at the vertex $v$?

**Definition 5.2** (DP-State). The following invariant should always hold:

$$DP[v][k] =$$
$$\max \left\{ f \mid \exists L, |L| \le k : \lambda_{(T_v)_{L,\tau,f}}(\{s\}, \{t\}) = n_v \cdot \tau + f \right\}$$

Where $n_v = |\mathcal{L}_{T_v}|$.
If that set is empty, we define $DP[v][k] = -\infty$.

We give the base cases and transitions, and later prove correctness via the invariant stated above.

**Definition 5.3.** We define an auxiliary "edge-bound" function for all edges $(u, v)$ in $G$.

$$\text{bound}_{(u,v)}(x) = \begin{cases} -\infty & x < -w(u,v) \\ w(u,v) & x > w(u,v) \\ x & \text{otherwise} \end{cases}$$

**Definition 5.4.** We define the following base cases and transitions:

- Base cases for leaves $l \in \mathcal{L}_T$:

$$DP[l][k] = \begin{cases} -\tau & \text{for } k = 0 \\ \infty & \text{for } k > 0 \end{cases}$$

- Transition for vertices $v \in V$ with two children $c_1, c_2$:

$$DP[v][k] = \max_{0 \le a \le k} \left( \begin{array}{c} \text{bound}_{(v,c_1)}(DP[c_1][a]) \\ + \text{bound}_{(v,c_2)}(DP[c_2][k-a]) \end{array} \right)$$

Now we show a sort of monotony of the DP-state invariant. Then we show, by induction on the binary tree, that the base cases and the transitions preserve the invariant as described in Definition 5.2.

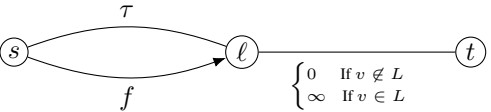

Figure 3. The structure of $(T_l)_{L,\tau,f}$ for some leaf $l$.

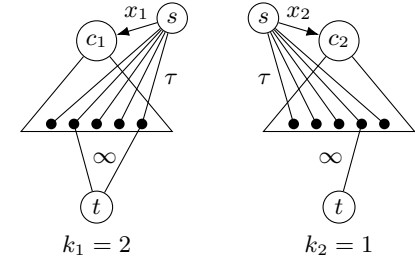

Figure 4. The structure of $(T_{c_1})_{L,\tau,f}$ and $(T_{c_2})_{L,\tau,f}$ for the children $c_1, c_2$ of $v$.

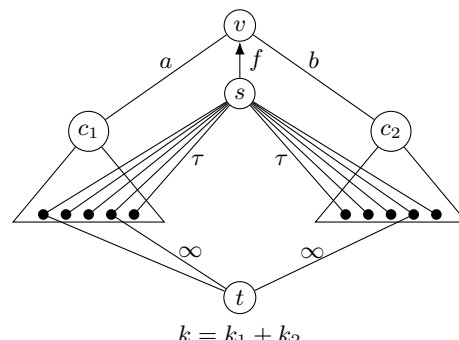

Figure 5. The structure of $(T_v)_{L,\tau,f}$

**Lemma 5.5.** *Given that* $DP[v][k] = f \ge 0$ *at some vertex* $v$ *and budget* $k$, *we have that* $\lambda_{(T_v)_{L,\tau,f'}}(\{s\}, \{t\}) = n_v \cdot \tau + f'$ *for all* $f'$ *with* $0 \le f' \le f$.

*Proof.* The prove is by induction. The base cases are the leaves. For some leaf $l \in \mathcal{L}_T$ we have $DP[l][0] = -\tau$ and $DP[l][1] = \infty$ by definition. The lemma does not cover the first case, so we only need to show it for the second case. We can look at Figure 3 and see that for any $f' \in \mathbb{R}$ there is a $s$-$t$ flow of value $\tau + f'$.

For the general case, as depicted in Figure 5, we notice that we can reduce it to two smaller instances as depicted in

Figure 4. Note that $n_v = n_{c_1} + n_{c_2}$. Now we have two cases:

- $a, b \geq 0$: We scale down both $a$ and $b$ by multiplying with $f'/f$. This is allowed by the induction hypothesis.

- $a < 0, b \geq 0$ or $a \geq 0, b < 0$: We assume, without loss of generality, that $a < 0, b \geq 0$. We set the new $b' = b - (f - f')$. We have

$$
\begin{aligned}
f = a + b &\implies b = f - a > f \\
f' \geq 0 &\implies f - f' \leq f \\
&\implies b' = b - (f - f') > 0,
\end{aligned}
$$

so $b' \geq 0$, and we can apply the induction hypothesis.

Note that the case $a, b < 0$ is impossible because $f = a + b \geq 0$. $\qquad\square$

**Lemma 5.6.** *The DP calculated with base cases and transitions as described in Definition 5.4 satisfies the invariant Definition 5.2.*

*Proof.* We show this by induction on the binary tree. For the base case on a leaf $v$, we observe that $T_v$ is just a single vertex. That means $(T_v)_{L,\tau,f}$ looks as in Figure 3. If $k = 0$, this implies $v \notin L$, forcing $f = -\tau$. If $k \geq 1$, we chose $L = \{v\}$, allowing arbitrarily large $f$. This immediately gives the base cases:

$$
\mathrm{DP}[l][k] = \begin{cases} -\tau & \text{for } k = 0 \\ \infty & \text{for } k > 0 \end{cases}
$$

Now we regard the transitions for a vertex $v$ with two children $c_1, c_2$. We have budget $k$, and we can choose how to distribute that among the two children. The subgraphs of the two children are shown in Figure 4. Now we try to combine them, this is depicted in Figure 5. We get that $f = a + b$. For this to be a feasible solution, we need: $a \leq x_1, b \leq x_2$ and $|a| \leq w(v, c_1), |b| \leq w(v, c_2)$. So by using $a = \mathrm{bound}_{(v,c_1)}(x_1)$ and $b = \mathrm{bound}_{(v,c_2)}(x_2)$ and applying Lemma 5.5, we get feasibility of the DP solution. That is, the value calculated by the DP is always achievable, and the solution can be recovered using backtracking.

Now we need to show that the optimal solution can be reproduced with such a construction, and thus be found by the DP. We look at the optimal solution, it has some $a, b$ with $a + b = f$ and $|a| \leq w(v, c_1), |b| \leq w(v, c_2)$. Also, it must choose some $k_1, k_2$ with $k_1 + k_2 \leq k$. By the induction hypothesis, the calculated DP solution at both $c_1$ and $c_2$ with budget $k_1$ and $k_2$ respectively is optimal, so it must be at least that of the optimal flow we are looking for. This directly implies that the DP finds an optimal construction. $\qquad\square$

Now we discuss how the optimal solution for the sink selection problem can be recovered. For some $L$ to be a solution to the sink selection problem, we need that the $s$-$t$-maxflow in $T_{L,\tau}$ has value $n \cdot \tau$. The graphs $T_{L,\tau}$ and $T_{L,\tau,f}$ are equivalent for $f = 0$. The DP calculates $f$ given some $k$. Using the DP monotony (Lemma 5.5) and the correctness of the DP (Lemma 5.6), we get that for all $k$ where $\mathrm{DP}[\mathrm{root}(T)][k] \geq 0$ there is a solution $L$ with $|L| = k$. So we choose the smallest $k$ with $\mathrm{DP}[\mathrm{root}(T)][k] \geq 0$. The corresponding selection of vertices $L$ can be recovered using standard DP backtracking techniques.

**Runtime.** We observe that our dynamic program has $O(|V_T| \cdot k)$ states. Every transition can be computed in $O(k)$ steps. The tree cut sparsifier is a binary tree with $|\mathcal{L}_T| = |V_G| = n$ leaves, so it has $O(n)$ vertices. This gives a runtime of $O(n \cdot k^2)$ for the dynamic program. We need to run a binary search to find the optimal $\tau$. So the total runtime of the algorithm is $O(T_{\text{TREECUTSPARSIFIER}} + n \cdot k^2 \cdot \log(n \cdot W))$, where $W$ is the aspect ratio of the edge weights and $T_{\text{TREECUTSPARSIFIER}}$ is the time required to compute a tree cut sparsifier. Depending on this time, we get different runtime and approximation trade-offs. Using (Räcke & Shah, 2014) achieves total polynomial runtime with $\tilde{O}(\log^{1.5} n)$ approximation, yielding Theorem 1.1. Using (Agassy et al., 2025) achieves total $\tilde{O}(n \cdot k^2)$ runtime[4] with $\tilde{O}(\log^2 n)$ approximation.

## 6. Experiments

Our algorithm relies on tree cut sparsifiers, for which there currently are no open source implementations that we are aware of. However, they are usually constructed by repeatedly decomposing a graph along sparse cuts.[5] Therefore, we use sparse-cut heuristics like METIS (Karypis & Kumar, 1997) to build a hierarchical decomposition resembling a tree cut sparsifier. We have implemented multiple such heuristics to compare them against each other and against the algorithms from (Cohen-Addad et al., 2025; Cesa-Bianchi et al., 2010; Guillory & Bilmes, 2009) on various graphs. Algorithm 1 describes how to achieve such a hierarchical decomposition given an algorithm BISECT that cuts a graph into two parts.

We construct $T$ by first creating a root vertex $r$ which corresponds to the whole graph $G$. Then we find a sparse cut $(S, V \setminus S)$ in $G$ using one of the heuristics. Now we create two new vertices $a, b$ corresponding to the induced subgraphs $G[S]$ and $G[V \setminus S]$ respectively. We add edges $(r, a)$ and $(r, b)$ with weights equal to the weight of the cut $w(S, V \setminus S)$, and recursively build the tree until arriving at

---

[4]For the aspect ratio of the edge weights $W \in \mathrm{poly}(n)$.

[5]For obtaining state-of-the-art guarantees on worst-case instances, additional delicate properties are necessary.

leaves corresponding to single vertices in $G$.

Our choice of edge weights ensures that the resulting tree fulfills the lower bound property of a tree cut sparsifier: For all $A_G \subseteq V_G$ we have $w_G(A, V \setminus A) \leq \lambda_T(A, \mathcal{L}_T \setminus A)$.

---

**Algorithm 1** HIERARCHICALDECOMPOSITION

---

**Input:** Graph $G = (V, E, w)$
**Output:** Edgelist of Tree
If $|G| = 1$, **return** $\emptyset$
$A, B \leftarrow \text{BISECT}(G)$
$e_A \leftarrow (V, A, w_G(A, V \setminus A))$
$e_B \leftarrow (V, B, w_G(B, V \setminus B))$
$E_A = \text{HIERARCHICALDECOMPOSITION}(G[A])$
$E_B = \text{HIERARCHICALDECOMPOSITION}(G[B])$
**return** $\{e_A, e_B\} \cup E_A \cup E_B$

---

In the following we describe which sparsest cut heuristics we used: FIEDLER and METIS (Spielman & Teng, 2004; Karypis & Kumar, 1997).

**Sparsest cut via *Fiedler Vector*.** We calculate the Fiedler vector of the graph and run spectral sweep to choose a cut with low sparsity (Spielman & Teng, 2004). The Fiedler vector, or algebraic connectivity, is the Eigenvector corresponding to the second smallest Eigenvalue of a graph Laplacian.[6] See Algorithm 2 in Section C. We remark that the computation of the Fiedler vector could be performed on a GPU in a straightforward manner.

We also tested a slight adaptation of this algorithm which forces the bisection to be balanced. In this case the algorithm has an additional parameter $\beta$ which specifies how balanced the cut should be. If a cut $(A, B)$ is chosen with either $\min(|A|, |B|) \leq n \cdot \beta$ we disregard that cut and choose the next best one. This can give big runtime improvements at the expense of quality.[7]

**Sparsest cut with *METIS* (Karypis & Kumar, 1997).** We use METIS (Karypis & Kumar, 1997) to calculate bisections. We run the algorithm $O(\sqrt{n})$ times with different target partition weights. They are distributed geometrically between 1 and $n/2$. Then we choose the partition with the best sparsity. See Algorithm 3 in Section C. We point out that the METIS calls in this algorithm could be parallelized naively.

**Experimental Setup.**[8] We compare our algorithm to the algorithms from (Cohen-Addad et al., 2025; Cesa-Bianchi

---

[6]The smallest Eigenvalue is 0.

[7]This strategy ensures that the resulting tree cut sparsifier is balanced.

[8]We provide the code used for running the experiments on GitHub: https://github.com/josia-john/icml2026-graph-label-selection

et al., 2010; Guillory & Bilmes, 2009). For nondeterministic algorithms we ran the experiment 10 times and show the standard deviation as a shaded area. We evaluate our algorithm with the following bisect algorithms/sparse cut heuristics:

> FIEDLER
>
> FIEDLERBALANCED with $\beta \in \{0.01, 0.1\}$
>
> METIS with #samples $\in \{\sqrt{n}, 10\sqrt{n}, 100\sqrt{n}\}$

We test our algorithms on various real world graphs from the Stanford Network Analysis Project (SNAP) (Leskovec & Krevl, 2014):

> ca-GrQc $\qquad |V| = \quad 4\,158 \quad |E| = \quad 13\,428$
> (Leskovec et al., 2007)
> ca-CondMat $\quad |V| = 21\,363 \quad |E| = \quad 91\,342$
> (Leskovec et al., 2007)
> ego-facebook $\quad |V| = \quad 4\,039 \quad |E| = \quad 88\,234$
> (McAuley & Leskovec, 2012)
> com-dblp $\qquad |V| = 317\,080 \quad |E| = 1\,049\,866$
> (Yang & Leskovec, 2012)

Additionally we tested on the Davis Southern Women Graph ($|V| = 32, |E| = 89$) for interpretable results on a small graph (Davis et al., 1941).

The graphs were preprocessed by choosing the largest connected component and deleting self-loops.

The experiments were run on a *AMD Ryzen Threadripper PRO 7955WX* processor (16 cores / 32 threads, up to 5.3 GHz). The different algorithms have very different runtime scaling behavior. Due to this we could not run all algorithms on all graphs, especially for large $k$. Therefore, some algorithms have more data points than others. For a runtime comparison see Table 1.

*Table 1.* Runtime (seconds) of all algorithms on ca-GrQc for $k \in \{10, 50, 100\}$. Real time denotes wall-clock time; user time denotes CPU time spent in user mode. The runtimes should be interpreted with caution: the experiments were not conducted in a fully controlled environment (e.g., multiple processes were running concurrently), and the implementations were not tuned for performance. Nevertheless, the table provides a coarse comparison of orders of magnitude and relative trends across methods.

| Budget | time | $k = 10$ | $k = 50$ | $k = 100$ |
|---|---|---|---|---|
| Guillory Bilmes | real | 144 | 842 | 1835 |
| | user | 645 | 1329 | 2303 |
| Cesa-Bianchi et al. | real | 14 | 8 | 13 |
| | user | 13 | 8 | 12 |
| Cohen-Addad et al. | real | 4967 | 15586 | 22956 |
| | user | 67808 | 222014 | 432845 |
| Ours FIEDLER | real | 22 | 21 | 22 |
| | user | 31 | 30 | 31 |

**Results.** Both FIEDLER and METIS perform well overall, with FIEDLERBALANCED running a lot faster at the cost of

quality. Figures 6 and 7 show the comparison. We achieve similar quality to current state of the art (Cohen-Addad et al., 2025). But our algorithm runs magnitudes faster. It has no problem running for many different values $k$ on a graph like ca-CondMat, see Figure 7, while other algorithms struggle even for just one value $k$ on smaller graphs like ca-GrQc.

We remark that the quality often remains constant for a sequence of budgets because there are numerous disjoint sparse cut with equal sparsity (For example degree one vertices).

Our algorithm can handle quite large graphs. Using METIS with #samples $= \sqrt{n}$ the algorithm runs within a few hours on com-dblp. We present those results in Table 2. For more results refer to Section C.

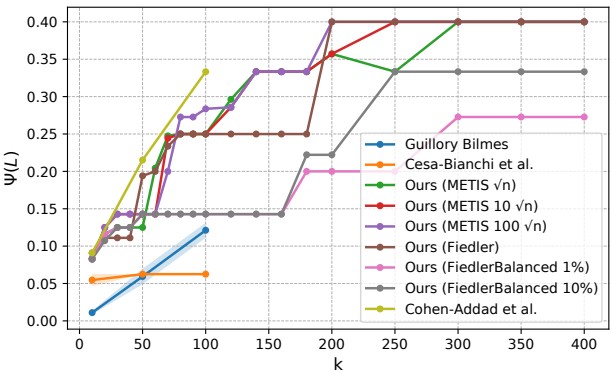

*Figure 6.* Comparison of all algorithms on ca-GrQc. Due to runtime constraints, we could not run all algorithms for every value of $k$.

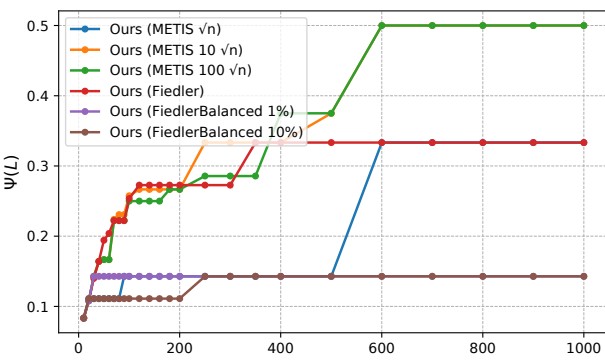

*Figure 7.* Comparison of our algorithm using different BISECT algorithms on ca-CondMat.

*Table 2.* Performance of METIS on com-dblp.

| Budget | $k = 50$ | $k = 500$ | $k = 5000$ |
|---|---|---|---|
| Ours (METIS $\sqrt{n}$) | 0.030 | 0.048 | 0.083 |

## 7. Conclusion

We present the first approximation algorithm for the graph label selection problem on general graphs that does not rely on resource augmentation. We also describe how to adapt the algorithm to give an optimal solution for the graph label selection problem on weighted trees.

It remains an interesting open problem to scale algorithms to massive graphs. One of the main bottlenecks in our framework is the dynamic program. An approximate version of this dynamic program could be solvable in linear time.

## Acknowledgements

The research of Simon Meierhans and Maximilian Probst Gutenberg leading to these results has received funding from grant no. 200021 204787 of the Swiss National Science Foundation. Simon Meierhans is supported by a Google PhD Fellowship.

## Impact Statement

At present, our results are primarily theoretical in nature. In the future, the techniques presented in this paper could be used to improve the reasoning capabilities of large language models and should therefore be used with caution as models become more capable.

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

## A. Solving Graph Label Selection optimally on Weighted Trees

We show that we can solve Graph Label Selection optimally on weighted trees by constructing a perfect tree cut sparsifier for weighted trees. Then the result follows immediately from.

**Lemma A.1.** *Given a tree $G = (V, E, w)$. There is a 1 tree cut sparsifier for G.*

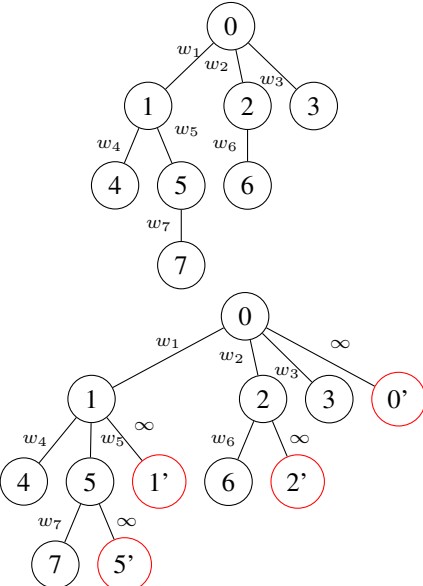

*Figure 8.* Transforming a tree $G$ (top) into a 1 tree cut sparsifier $T$ (bottom). We connect a new leaf with weight $\infty$ to every internal vertex in $G$.

*Proof.* Let $G = (V, E, w)$ be a weighted tree. We construct a tree $T = (V_T, E_T, w_T)$ as follows (see Figure 8). Start with $G$ and for every internal vertex $v \in V \setminus \mathcal{L}_G$ add a new leaf $v'$ and an edge $(v, v')$ with weight $\infty$.

Now we have a natural one to one mapping between the leaves $\mathcal{L}_T$ in $T$ and the vertices $V$ in $G$. For $v \in \mathcal{L}_T$ it corresponds to $v \in V$, and for $v' \in \mathcal{L}_T$ it corresponds to $v \in V$.

We show that given any $A \subseteq V$ and its mapping to the leaves of the tree cut sparsifier $A' \subseteq \mathcal{L}_T$ with

$$A' = \{v' \mid v \in A, v' \in V_T\} \cup \{v \mid v \in A, v' \notin V_T\}$$

the following holds:

$$\lambda_T(A', \mathcal{L}_T \setminus A') = w_G(A, V \setminus A),$$

which implies that $T$ is a 1 tree cut sparsifier according to Definition 3.1.

Let us analyze the set $S$ materializing the mincut

$$\lambda_T(A', \mathcal{L}_T \setminus A') = \min_{A' \subseteq S \subseteq V_T \setminus (\mathcal{L}_T \setminus A')} w_T(S, V_T \setminus S).$$

For any vertex $v \in A'$ we must, by definition, have $v \in S$. For every vertex $v \in \mathcal{L}_T \setminus A'$ we must, by definition, have $v \notin S$. This leaves us with internal vertices $v \in V_T \setminus \mathcal{L}_T$. By our construction, every internal vertex $v$ has a leaf $v'$, for which we know whether it is in $S$ or not. Assuming $v' \in S$, if we would choose $v \notin S$, then the edge $(v, v')$ with weight $\infty$ would contribute to the cut size, making it $\infty$ too. So we must choose $v \in S$. If $v' \notin S$, then $v \notin S$ by the same argument.

So we must choose $S = A \cup A'$. All $\infty$ edges are not contributing, so the cut is equivalent to the cut $(A, V \setminus A)$ in $G$, giving

$$\lambda_T(A', \mathcal{L}_T \setminus A') = w_T(S, V_T \setminus S) = w_G(A, V \setminus A),$$

concluding our proof. $\square$

## B. Generalizing to Vertex Importance

In this article we have described how to maximize the function $\Psi(L) = \min_{C \subseteq V \setminus L} w(C, V \setminus C)/|C|$. But our algorithm can easily be extended to allow for maximization of any function of the form

$$\min_{C \subseteq V \setminus L} \frac{w(C, V \setminus C)}{\sum_{c \in C} f(c)},$$

where $f : V \to \mathbb{R}_{\geq 0}$ is an "importance" function of vertices. The standard label selection problem tries to maximize a sparsity metric. That would be $f(v) = 1$ for all $v \in V$. This extension allows us to optimize for many more metrics, with one important example being a conductance like metric:

$$\min_{C \subseteq V \setminus L} \frac{w(C, V \setminus C)}{\mathrm{vol}(C)}$$

Here we use the fact that $\mathrm{vol}(C) = \sum_{c \in C} \deg(c)$.

The algorithm is almost the same as we have described above, the main difference is just the base cases of the dynamic program. Here we give the slightly modified proofs.

We start by defining the new objective function, for both GLS and LLS.

**Definition B.1** (Graph Label Selection Problem with Vertex Importance)**.** Given a graph $G = (V, E, w)$, a vertex importance function $f : V \to \mathbb{R}$, and a budget $k \in \mathbb{N}$, find a set $L \subset V, |L| \leq k$ that maximizes the objective

$$\Psi_G^f(L) \coloneqq \min_{C \subseteq V \setminus L} \frac{w(C, V \setminus C)}{\sum_{c \in C} f(c)}$$

**Definition B.2** (Leaf Label Selection Problem with Vertex Importance)**.** Given a tree $T = (V, E, w)$ and a budget $k \in \mathbb{N}$, find a set $L \subseteq \mathcal{L}_T, |L| \leq k$ that maximizes the objective

$$\widehat{\Psi}_T^f(L) = \min_{C \subseteq \mathcal{L}_T \setminus L} \frac{\lambda_T(C, \mathcal{L}_T \setminus C)}{\sum_{c \in C} f(c)}$$

In the following we describe how all three steps of our algorithm can be adapted to work with vertex importance.

**Reducing to Binary Tree.** Given is the graph $G = (V, E, w)$. From Lemma 3.6 we get that we have some binary $\alpha$ tree cut sparsifier $T = (V_T, E_T, w)$. Now we need to show Lemma 3.4 with respect to the new importance function $f : V \to \mathbb{R}$.

**Lemma B.3.** *Given a graph $G = (V_G, E_G, w_G)$ and a corresponding $\alpha$ tree cut sparsifier $T = (V_T, E_T, w_T)$. For any solution $L \subseteq \mathcal{L}_T = V$ we have:*

$$\widehat{\Psi}_T^f(L) \leq \alpha \Psi_G^f(L)$$

*Proof.*

$$\begin{aligned}
\widehat{\Psi}_T^f(L) &= \min_{C \subseteq \mathcal{L}_T \setminus L} \frac{\lambda_T(C, \mathcal{L}_T \setminus C)}{\sum_{c \in C} f(c)} \\
&\leq \min_{C \subseteq V \setminus L} \frac{\alpha \cdot w_G(C, V \setminus C)}{\sum_{c \in C} f(c)} \\
&= \alpha \Psi_G^f(L)
\end{aligned}$$

$\square$

**Lemma B.4.** *Given a graph $G = (V_G, E_G, w_G)$ and a corresponding $\alpha$ tree cut sparsifier $T = (V_T, E_T, w_T)$. For any solution $L \subseteq V = \mathcal{L}_T$ we have:*

$$\widehat{\Psi}_T^f(L) \geq \Psi_G^f(L)$$

*Proof.*

$$\begin{aligned}
\widehat{\Psi}_T^f(L) &= \min_{C \subseteq \mathcal{L}_T \setminus L} \frac{\lambda_T(C, \mathcal{L}_T \setminus C)}{\sum_{c \in C} f(c)} \\
&\geq \min_{C \subseteq V \setminus L} \frac{w_G(C, V \setminus C)}{\sum_{c \in C} f(c)} \\
&= \Psi_G^f(L)
\end{aligned}$$

$\square$

**Corollary B.5.** *Given a graph $G = (V, E, w)$, a vertex importance function $f$, and a corresponding $\alpha$ tree cut sparsifier $T = (V_T, E_T, w_T)$. Let $L'$ be the optimal solution of LLS on $T$. Then $L'$ is a solution for GLS with approximation factor $\alpha$.*

*Proof.* Let $L^*$ be the optimal solution of GLS on $G$. Using Lemmas B.3 and B.4, and optimality of $L'$ for LLS, we get:

$$\alpha \Psi^f(L') \geq \widehat{\Psi}_T^f(L')$$
$$\widehat{\Psi}_T^f(L') \geq \widehat{\Psi}_T^f(L^*) \geq \Psi_G^f(L^*)$$

$\square$

**Reducing to Flow Problem.** We define a new flow graph, with respect to the vertex importance function $f$:

**Definition B.6** (Flow Graph with Vertex Importance)**.** Given a tree $T = (V, E, w)$, a vertex importance function $f : V \to \mathbb{R}$, a set $L \subseteq \mathcal{L}_T$, and a threshold $\tau \in \mathbb{R}$. We construct $T_{L,\tau}^f$:

- Vertex Set: $V \cup \{s, t\}$

- Copy: Every edge $e \in E$ is also present in $T_{L,\tau}^f$ with the same weight.

- Source: For every leaf $v \in \mathcal{L}_T$, there is an edge $(s, v)$ with weight $\tau \cdot f(v)$ in $T_{L,\tau}^f$.

- Sink: For every chosen leaf $v \in L$, there is an edge $(v, t)$ with weight $\infty$ in $T_{L,\tau}^f$.

**Lemma B.7.** *If $\lambda_{T_{L,\tau}^f}(\{s\}, \{t\}) < \tau \cdot \sum_{v \in \mathcal{L}_T} f(v)$, then $\widehat{\Psi}_T^f(L) < \tau$.*

*Proof.* By the definition of mincut, there is some $S'$ such that:

$$\lambda_{T_{L,\tau}^f}(\{s\}, \{t\}) = w_{T_{L,\tau}^f}(S' \cup \{s\}, (V \setminus S') \cup \{t\})$$

We notice that $S', L$ must be disjoint because all vertices in $L$ have an edge of infinite capacity to $t$. We can write the size of this cut as the sum of two parts: The cut in the original tree $T$ plus the edges from the new sink $s$ to vertices in $(V \setminus S') \cup \{t\}$:

$$\begin{aligned}
&w_{T_{L,\tau}^f}(S' \cup \{s\}, (V \setminus S') \cup \{t\}) \\
&= w_T(S', V \setminus S') + w_{T_{L,\tau}^f}(\{s\}, (V \setminus S') \cup \{t\})
\end{aligned}$$

The second part of the sum is just the edges from $\{s\}$ to leaves $v \in \mathcal{L}_T \setminus S'$. The edge $(s, v)$ has weight $\tau \cdot f(v)$, so the sum can be written as:

$$\begin{aligned}
&w_{T_{L,\tau}^f}(S' \cup \{s\}, (V \setminus S') \cup \{t\}) \\
&= w_T(S', V \setminus S') + \tau \cdot \sum_{v \in \mathcal{L}_T \setminus S'} f(v)
\end{aligned}$$

Now we have

$$\lambda_{T_{L,\tau}^f}(\{s\}, \{t\}) = w_T(S', V \setminus S') + \tau \cdot \sum_{v \in \mathcal{L}_T \setminus S'} f(v).$$

Plugging this into the LHS of the statement $\lambda_{T_{L,\tau}^f}(\{s\}, \{t\}) < \tau \cdot \sum_{v \in \mathcal{L}_T} f(v)$ we get:

$$w_T(S', V \setminus S') + \tau \cdot \sum_{v \in \mathcal{L}_T \setminus S'} f(v) < \tau \cdot \sum_{v \in \mathcal{L}_T} f(v)$$

$$\implies \quad w_T(S', V \setminus S') < \tau \cdot \sum_{v \in S' \cap \mathcal{L}_T} f(v)$$

Let us construct $S = S' \cap \mathcal{L}_T$. We have

$$\lambda_T(S, \mathcal{L}_T \setminus S) \leq w_T(S', V \setminus S') < \tau \cdot \sum_{v \in S} f(v)$$

Because $S, L$ are disjoint, this gives

$$\widehat{\Psi}_T^f(L) = \min_{C \subseteq \mathcal{L}_T \setminus L} \frac{\lambda_T(C, \mathcal{L}_T \setminus C)}{\sum_{c \in C} f(c)}$$

$$< \frac{\tau \cdot \sum_{v \in S} f(v)}{\sum_{v \in S} f(v)} = \tau$$

$\square$

**Lemma B.8.** *If* $\widehat{\Psi}_T^f(L) < \tau$, *then* $\lambda_{T_{L,\tau}^f}(\{s\}, \{t\}) < \tau \cdot \sum_{v \in \mathcal{L}_T} f(v)$

*Proof.* By the definition of $\widehat{\Psi}^f$, there is some $S$ disjoint from $L$ such that:

$$\widehat{\Psi}_T^f(L) = \frac{\lambda_T(S, \mathcal{L}_T \setminus S)}{\sum_{v \in S} f(v)}$$

$$\implies \qquad \lambda_T(S, \mathcal{L}_T \setminus S) < \tau \cdot \sum_{v \in S} f(v)$$

Now, by the definition of mincut, there is some $S'$ with $S \subseteq S' \subseteq V \setminus (\mathcal{L}_T \setminus S)$ such that:

$$w_T(S', V \setminus S') < \tau \cdot \sum_{v \in S} f(v)$$

We analyze $w_{T_{L,\tau}^f}(S' \cup \{s\}, (V \setminus S') \cup \{t\})$. As above, we can split this up:

$$w_{T_{L,\tau}^f}(S' \cup \{s\}, (V \setminus S') \cup \{t\})$$
$$= w_T(S, V \setminus S) + w_{T_{L,\tau}^f}(\{s\}, (V \setminus S') \cup \{t\})$$

We can bound $w_{T_{L,\tau}}(\{s\}, (V \setminus S') \cup \{t\})$ just as above. It is the edges from $s$ to leaves $\mathcal{L}_T \setminus S'$. That gives

$$w_{T_{L,\tau}}(S' \cup \{s\}, (V \setminus S') \cup \{t\})$$
$$= w_T(S', V \setminus S') + \tau \cdot \sum_{v \in \mathcal{L}_T \setminus S'} f(v)$$

$$< \tau \cdot \sum_{v \in S} f(v) + \tau \cdot \sum_{v \in \mathcal{L}_T \setminus S'} f(v) \leq \tau \cdot \sum_{v \in \mathcal{L}_T} f(v)$$

This directly implies $\lambda_{T_{L,\tau}^f}(\{s\}, \{t\}) < \tau \cdot \sum_{v \in \mathcal{L}_T} f(v)$.

$\square$

**Corollary B.9.**

$$\widehat{\Psi}_T^f(L) \geq \tau \iff \lambda_{T_{L,\tau}^f}(\{s\}, \{t\}) = \tau \cdot \sum_{v \in \mathcal{L}_T} f(v)$$

*Proof.* We have $\lambda_{T_{L,\tau}^f}(\{s\}, \{t\}) \leq \tau \cdot \sum_{v \in \mathcal{L}_T} f(v)$ because the degree of $s$ is $\tau \cdot \sum_{v \in \mathcal{L}_T} f(v)$. Then we just apply Lemmas B.7 and B.8. $\square$

**Definition B.10** (Sink Selection Problem)**.** Given a tree $T = (V, E, w)$ and a parameter $\tau$, find $L \subseteq \mathcal{L}_T$ with minimal $|L|$, such that the $s$-$t$-maxflow in $T_{L,\tau}^f$ has value $\tau \cdot \sum_{v \in \mathcal{L}_T} f(v)$.

The monotony from Lemma 4.6 holds just as in Section 4, so we can again find $L$ using binary search.

**Solving the Flow Problem using Dynamic Programming.**
This part is pretty immediate. We can use the same DP-state as in Section 5, just using the new flow graph $T_{L,\tau}^f$. The only thing that changes is the base cases for leaves. Instead of $-\tau$, we now use $-\tau \cdot f(v)$ at the leaf $v$. It's easy to see that this base case agrees with the new DP-state. The proof that the transitions are correct still works here.

# C. Further Experiments

---

**Algorithm 2** BISECT (FIEDLER)

---

**Input:** Graph $G = (V, E, w)$ and its Laplacian $L$
**Output:** Bisection $(A, B)$ of $V$

$\vec{v} \leftarrow \lambda_2(L)$        $\triangleright$ the second eigenvector of $L$
best $\leftarrow$ (score : $\infty$, $A$ : {}, $B$ : {})
**for** $t \in$ elements$(\vec{v})$ **do**     $\triangleright$ iterate over all values in $\vec{v}$
    $A \leftarrow \{i \mid \vec{v}_i \leq t\}$
    $B \leftarrow \{i \mid \vec{v}_i > t\}$
    score $\leftarrow \frac{w_G(A,B)}{\min(|A|,|B|)}$
    **if** score $<$ best.score **then**
        best $\leftarrow$ (score, $A$, $B$)
    **end if**
**end for**
**return** (best.$A$, best.$B$)

---

---

**Algorithm 3** BISECT (METIS)

---

**Input:** Graph $G = (V, E, w)$ and a number of #samples.
**Output:** Bisection $(A, B)$ of $V$

best $\leftarrow$ (score $: \infty, A : \{\}, B : \{\})$
**for** $w \in$ geomspace$(1, n/2, \#samples)$ **do**
    $(A, B) \leftarrow$ METIS$(G, \text{tpwgts} = [w/n, 1 - w/n])$
    score $\leftarrow \frac{w_G(A,B)}{\min(|A|,|B|)}$
    **if** score $<$ best.score **then**
        best $\leftarrow$ (score, $A, B$)
    **end if**
**end for**
**return** (best.$A$, best.$B$)

---

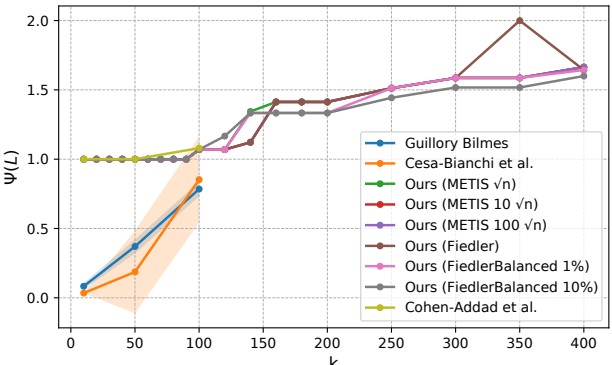

*Figure 9.* Comparison of all algorithms on ego-facebook. Due to runtime constraints, we could not run all algorithms for every value of $k$.

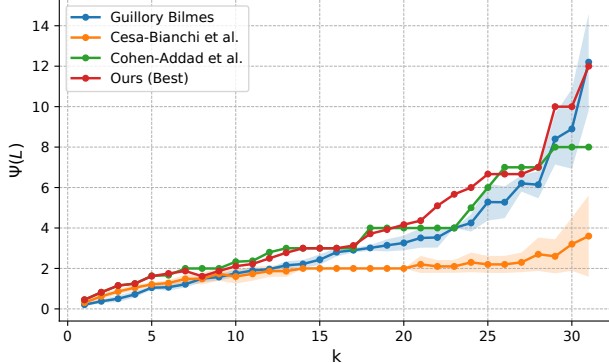

*Figure 10.* Comparison of all algorithms on the Davis Southern Women Graph. We include this figure to compare all algorithms for many different values of $k$. For "Ours (Best)" we ran our algorithm with all bisect algorithms and plotted the best result.

