# OpenReview forum: "An Approximation Algorithm for Graph Label Selection"
_ICML.cc/2026/Conference — ICML 2026 regular_

### Official Review · Reviewer_J6jD · 2026-02-28

**Soundness:** 3
**Presentation:** 3
**Significance:** 3
**Originality:** 3
**Overall Recommendation:** 4
**Confidence:** 3

**Summary:**

This paper studies the design of approximation algorithms for the NP-hard graph label selection (GLS) problem. The input of GLS problem consists of an $n$-vertex graph together with a budget parameter $k$. The goal is to select at most $k$ vertices whose labels enable accurate prediction of the labels on the remaining vertices, i.e., to achieve the optimal objective value $\textrm{OPT}_k$.

Previous work has provided an $\alpha$-resource approximation algorithm, which are allowed to use at most $\alpha\cdot k$ selected vertices while attaining the optimal value $\textrm{OPT}_k$. An alternative direction, which is pursued in this paper, is to restrict the algorithm to $k$ vertices (i.e., under the standard budget constraint) and instead approximate the objective value, namely to obtain a solution of value at least $\textrm{OPT}_k/\beta$, where $\beta$ denotes the approximation ratio.

The notion of $\beta$-competitiveness w.r.t. $\textrm{OPT}_k$ was previously posed as an open problem (Cohen-Addad et al., 2025). The authors claim to be the first to resolve this question, presenting an algorithm achieving an approximation ratio of $\beta=\tilde{O}(\log^{1.5} n)$. Moreover, this paper conducted experiments to validate its theoretical results.

**Compliance With Llm Reviewing Policy:**

Affirmed.

**Key Questions For Authors:**

**Questions:**
* It seems that the approximation ratio of GLS comes from the approximation ratio for tree cut sparsifier. If this is the case, if there exists an $O(\log n)$ tree cut sparsifier, can we obtain an $O(\log n)$ approximation ratio for GLS? Previous work proved that the lower bound of tree cut sparsifier is $\Omega(\log n)$, so does GLS under the standard budget constraint have an $\Omega(\log n)$ or better lower bound for the approximation ratio?
* Could the DP be reformulated as a min-cost flow or convex optimization on trees?

* About the experiments,
    * since the true $\textrm{OPT}_k$ is unknown due to NP-hardness, how should we interpret the reported objective values? Have the authors attempted exact computation on small graphs or derived upper/lower bounds for reference?
    * I am also curious about the runtime of the overall algorithm for different values of $k$ and different graph sizes. Could the authors report these results?


**Typos:**
* lines 112, 120, 126, 138 and many other places: missing a period after the equation.

**Limitations:**

yes

**Strengths And Weaknesses:**

**Strengths:**
* This paper is the first to give an approximation algorithm for GLS under the standard budget constraint, thus resolving an open question proposed in (Cohen-Addad et al., 2025).
* Technically, there is an elegant reduction:

    * firstly, reduce GLS to solve the LLS problem on a binary tree,
    * then, reduce LLS to solve a sink selection problem (flow problem),
    * finally, use DP to exactly solve the flow problem on binary tree.

* About experiments, the author conducted experiments on various real-world graph datasets of different sizes and compared the results with those from previous studies.
* The writing logic of this paper is clear and coherent.

**Weaknesses:**
* DP results in a $O(nk^2)$ time, which might be unacceptable when $k\in \Theta(n)$.
* While acceptable for proof-of-concept, the experimental section is not deeply tied to the theoretical result (see also questions below).

---

> ### Author Rebuttal · Authors · 2026-03-27
>
> We thank the reviewer for their work reviewing our paper. We would like to respond to the questions they raised.
>
> 1. The reviewer is correct to point out that the approximation ratio is entirely due to the approximation ratio of the tree cut sparsifer, and that tree-cut sparsifiers cannot be improved beyond $O(\log n)$. Konstantin Andreev et al. (2009) [3] show a lower bound of $\Omega(\log n)$ for simultaneous source location. This translates to an $\Omega(\log n)$ lower bound for resource augmented approximation and we suspect that this might be related to a lower bound on proper approximation. Currently however, there is only a constant lower bound on the approximation of GLS, and we believe that closing this gap is an interesting direction for future work.
>
> 2. The problem can be re-formulated as a sub-modular optimization problem over trees. We are not aware of a formulation in terms of minimum-cost flow.
>
> 3. We did not attempt to solve the problem exactly to show how far we are from OPT, and believe that it is difficult to do so on all instances that are meaningful. We report running times in Table 2 in the appendix of our article.
>
> [3] Simultaneous source location
> Konstantin Andreev, Charles Garrod, Daniel Golovin, Bruce Maggs, and Adam Meyerson
> ACM Transactions on Algorithms

---

> > ### Author Rebuttal · Reviewer_J6jD · 2026-04-03
> >
> > I have no other questions.

---

### Official Review · Reviewer_JYUJ · 2026-03-03

**Soundness:** 2
**Presentation:** 2
**Significance:** 2
**Originality:** 3
**Overall Recommendation:** 2
**Confidence:** 4

**Summary:**

The author focuses on the graph labeling problem, which requires selecting k vertices as a label set under a given budget constraint k, such that the labels of the remaining vertices can be accurately predicted. This essentially involves extracting a representative subset from the entire graph. Addressing the shortcomings of existing approaches—which either rely on resource expansion (using far more labeled vertices than k) or offer heuristics with no provable guarantees—this paper introduces the first O~(log1.5n) approximation algorithm that requires no resource expansion. The algorithm achieves this through two transformations (converting the general graph problem into a binary tree problem, then into a flow problem), ultimately solving it via dynamic programming with theoretically guaranteed approximation performance. Experimental validation demonstrates that the heuristic variant of this algorithm achieves significantly improved scalability compared to existing methods while maintaining comparable performance, enabling it to handle larger-scale graphs. The authors aim to present a novel approach to graph label selection that combines theoretical guarantees with practical value. Overall, this paper explores a highly relevant and important topic in graph learning, offering significant reference value for fields such as active learning and dataset distillation.

**Compliance With Llm Reviewing Policy:**

Affirmed.

**Key Questions For Authors:**

- The absence of an open-source implementation for tree-cut sparsification may impact algorithm reproducibility. How do the heuristic sparse-cut method and theoretical tree-cut sparsification differ in performance (approximation ratio, runtime)? Are there plans to supplement a complete implementation of theoretical tree-cut sparsification?
- Why were emerging methods such as active learning based on graph neural networks and diversity sampling algorithms not included in the comparative baseline? Could these comparisons be added to more clearly highlight the core advantages of this algorithm in theoretical guarantees and practical performance?
- The algorithm's time complexity is O(n⋅k²). Does efficiency significantly decrease when k is large (e.g., k=1000) or in ultra-large-scale graphs (hundreds of millions of nodes)? Are there targeted lightweight optimization approaches?
- What are the primary limitations of this algorithm? For instance, is it sensitive to edge weight distributions? Is it applicable to dynamic graphs (where topology changes over time)? Does the approximation ratio exhibit noticeable fluctuations between dense and sparse graphs? Please discuss these issues and potential directions for improvement.

**Limitations:**

The authors did not explicitly discuss the limitations of this study. Potential limitations of the algorithm include: 1) Core reliance on tree-cut sparsification techniques, with existing open-source implementations lacking, making direct verification of theoretical approximation ratios difficult; heuristic alternatives may result in partial performance loss; 2) The dynamic programming component has a time complexity of O(n⋅k²), which may be inefficient for large k values or extremely large graphs; 3) Sensitivity to edge weight distribution, where approximation performance may fluctuate in graphs with highly disparate edge weights; 4) It is not adapted for dynamic graph scenarios; when the topological structure changes, the entire process must be re-executed, lacking an incremental update mechanism; 5) Its adaptability in complex scenarios such as multi-label and heterogeneous graphs has not been verified in experiments. These limitations require improvement in subsequent research through optimizing tree cut sparsification implementation, introducing incremental dynamic programming, and designing adaptive edge weight processing mechanisms.

**Strengths And Weaknesses:**

Strengths

- This work resolves the long-standing open problem of “approximation algorithms without resource expansion under fixed budgets” in graph label selection, filling a theoretical gap in the field. The problem holds broad application prospects in active learning, context selection for LLM inference, and dataset distillation, demonstrating significant academic and practical value.
- Testing on multiple real-world graph datasets (e.g., ca-GrQc, com-dblp) demonstrated comparable performance to existing mainstream heuristic algorithms while achieving significantly enhanced scalability. The algorithm efficiently processes large-scale graphs with over 300,000 nodes, validating its practical potential.
- The algorithm seamlessly extends to scenarios involving vertex importance, supports optimization of various objective functions (e.g., conductivity metrics), and offers broad applicability, providing flexible solutions for diverse practical needs.

Weaknesses
- The algorithm's core relies on tree-cut sparsification techniques, but currently lacks open-source implementations. In experiments, a sparse-cut heuristic was used as a substitute, which may affect the actual realization of the theoretical approximation ratio. The performance difference between the heuristic and the theoretical algorithm was not explicitly clarified.
- The study excludes emerging graph sampling or representative subset selection algorithms (e.g., active learning methods based on graph neural networks, diversity sampling algorithms), making it difficult to fully highlight this approximation algorithm's unique advantages in complex graph scenarios.
- Although experiments handled graphs with 300,000 nodes, the paper does not explore the algorithm's performance on ultra-large-scale graphs with hundreds of millions of nodes. The dynamic programming component has a time complexity of O(n⋅k²), which may face efficiency bottlenecks when k is large.
- The paper omits discussion of inherent limitations, such as sensitivity to edge weight distributions, adaptability in dynamic graph scenarios, and approximation ratio fluctuations across different graph structures (e.g., dense vs. sparse graphs). This omission hinders future research improvements and extensions.

---

> ### Author Rebuttal · Authors · 2026-03-27
>
> We thank the reviewer for their work reviewing our paper. We would like to respond to the questions they raised.
>
> 1. We only use publicly available heuristics for generating the tree, and plan to release the code used to obtain our results. This ensures reproducability of our results. We do not currently plan to produce a theoretically accurate implementation of tree-cut sparsifiers. Although theoretically fast, these algorithms are not yet at a stage where their performance translates to practice. Furthermore, the approximation gurarantees for fast algorithms hide constant factors that make it dubious if they would even improve on practical heuristics. Real-world graphs are typically far from worst case instances, and we believe that this fact should be exploited for building fast and practical algorithms.
>
> 2. These novel developments are very interesting. Since they optimize for different objectives, we feel that it is difficult to make a fair comparison with our method. We therefore limit our experimental evaluation to the algorithms that optimize the GLS objective and are considered in prior work.
>
> 3. There is no simple way to improve the dependence on $k$ in our formulation. Computing the state function in a bottom up manner is equivalent to a (max, +) convolution, and conditional lower bounds from fine grained complexity suggest that this cannot be improved substantially beyond $k^{2 - \epsilon}$. In the special case where all the entries are integers in $O(k)$, this can be improved to $k^{1.5}$ via a recent result by Chi et al. (2022) [1]. In particular, this captures integer weighted graphs without vertex importances. Furthermore, when $k$ is very large, i.e. $k = \Omega(n)$, and the tree-cut sparsifier is balanced, then the overhead is not $n^3$ but just $n^2$. We point out that in theory, we obtain balanced tree cut sparsifiers, and the practical heuristics are typically balanced, too.
> In practice, the overhead could be reduced by not evaluating every single possible DP state, but instead appealing to monotonicity and periodical evaluation. This would significantly speed up the algorithm, but destroys the theoretical guarantees we are after. We also want to point out that the runtime dependence on $k^2$ comes from the evaluation of the DP state, which, if implemented carefully, is very efficient.
>
> 4. Without vertex importances, the GLS objective is quite sensitive to outlier vertices. We therefore suggest using vertex importances that are depending on the degrees of the vertices for more stable results. Otherwise, the algorithm is not very sensitive to edge weight changes.
> A tree-cut sparsifier can be maintained efficiently in a fully dynamic graph via an algorithm of van den Brand et al. (2024) [2]. We believe that it is difficult to bound the number of DP updates that such changes in the structure of the tree cause, but exploring practical version of such a dynamic algorithm is interesting future work.
>
>
> [1]: Faster Min-Plus Product for Monotone Instances
> Shucheng Chi, Ran Duan, Tianle Xie, Tianyi Zhang
> STOC 2022
>
> [2] Almost-Linear Time Algorithms for Decremental Graphs: Min-Cost Flow and More via Duality
> Jan van den Brand, Li Chen, Rasmus Kyng, Yang P. Liu, Simon Meierhans, Maximilian Probst Gutenberg, Sushant Sachdeva
> STOC 2024

---

> > ### Author Rebuttal · Reviewer_JYUJ · 2026-04-05
> >
> > 1. It is suggested to supplement comparative experiments between heuristic methods and theoretical tree-cut sparsification algorithms on real-world graphs, and further support the core conclusion that “theoretical algorithms have large constant factors and their practical performance is inferior to heuristics” through quantitative results such as running time and sparsification quality.
> >
> > 2. Regarding the sensitivity of the GLS objective to outlier vertices, it is recommended to expand the design of vertex importance. In addition to vertex degrees, topological features such as centrality can be incorporated, and the improvement of result stability brought by different weighting schemes should be verified through experiments.
> >
> > 3. It is suggested to clarify the practical feasibility in dynamic graph scenarios in the paper: analyze the impact of tree structure changes on DP updates, or provide a simplified framework for the dynamic maintenance version, so as to make the direction of future work more concrete and instructive.

---

> > > ### Author Response · Authors · 2026-04-07
> > >
> > > 1. For unweighted graphs, such as the ones we experiment with, the resource augmented algorithm of Cohen-Addad et al. which inherits the constant of sub-modular function maximization is provably obtaining a logarithmic overhead with a small constant on average. Our algorithm essentially matches this algorithm, which shows that the heuristics are achieving high quality trees in general. While it would be interesting to compare our algorithm to tree-cut sparsifiers with theoretical guarantees, we are not aware of any implementation of such an algorithm. Further, we believe that the quality of a theoretical algorithm is also sensitive to implementation details.
> > > 2. This article focuses on presenting a new algorithmic approach to the GLS optimization problem with better theoretical guarantees. The properties of the GLS objective itself, such as its sensitivity to outlier vertices, are out of scope for our article.
> > > 3. In theory, tree-cut sparsifiers can be maintained dynamically with sub-polynomial approximation and update time [1]. Since these have low depth, it is not out of the question that the dynamic program can be maintained, too. If the orientation (i.e. the direction to the root) for each edge in the tree would stay fixed, we could simply update all the edges on the path to the root. Since the number of re-orientations in a bounded depth tree cannot be too large, we believe that such an algorithm is plausible.
> > >
> > > [1] Almost-Linear Time Algorithms for Decremental Graphs: Min-Cost Flow and More via Duality
> > > Jan van den Brand, Li Chen, Rasmus Kyng, Yang P. Liu, Simon Meierhans, Maximilian Probst Gutenberg, Sushant Sachdeva

---

### Official Review · Reviewer_WgKz · 2026-03-10

**Soundness:** 3
**Presentation:** 3
**Significance:** 3
**Originality:** 3
**Overall Recommendation:** 4
**Confidence:** 3

**Summary:**

This paper gives the first approximation algorithm for graph label selection without resource augmentation that obtains an O(log^1.5 n)-approximation. The algorithm uses tree cut sparsifiers and a dynamic programming approach. It also provides an exact result on weighted trees and a natural extension to vertex-importance objectives.

**Compliance With Llm Reviewing Policy:**

Affirmed.

**Final Justification:**

My overall assessment of the paper is positive. I had some minor questions which the authors addressed. The problem appears to be very natural and interesting, but this is the first time I encounter it, so my score is 4 and not 5 mostly because it is hard for me to assess the true importance of the problem.

**Key Questions For Authors:**

- Can the authors clarify more explicitly which parts of the algorithm dominate the empirical performance?

- Can the quadratic dependence on k in the runtime be improved?

**Limitations:**

yes

**Strengths And Weaknesses:**

Strengths: Overall the paper considers a very natural and important problem. They provide SOTA theoretical results accompanied by extensive experiments. The paper addresses a natural open question left by prior work by giving a fixed-budget approximation algorithm for graph label selection, rather than relying on resource augmentation. I found the technical route reasonably elegant: the problem is reduced to a tree-based formulation via tree cut sparsifiers, then to a flow-style problem on a binary tree, and finally solved exactly on that tree using dynamic programming. Experimental results show that the suggested algorithm is much faster than existing algorithms. The paper is sound and well written.

Weaknesses:
- It looks like the quality of the results is below those of Cohen-Addad et al, and the main improvement is with regard to speed. I wonder if there are any natural improvements or optimizations that can be applied to the result of Cohen-Addad et al to make it faster? Did you try any optimizations?

- As I understand it, the experimental implementation is not a direct implementation of the theoretical algorithm because the tree-cut-sparsifier primitive is replaced by sparse-cut heuristics. This is perfectly reasonable as a proof of concept, but it means the experiments are only an indirect validation of the theoretical result.

---

> ### Author Rebuttal · Authors · 2026-03-27
>
> We thank the reviewer for their work reviewing our paper. We would like to respond to the questions they raised.
>
> 1. The performance depends on the heuristic used to obtain the tree. The evaluation of the DP state is very simple, and thus quite efficient. In our experiments the runtime of the DP was not the runtime bottleneck.
>
> 2. There is no simple way to improve the dependence on $k$ in our formulation. Computing the state function in a bottom up manner is equivalent to a (max, +) convolution, and conditional lower bounds from fine grained complexity suggest that this cannot be improved substantially beyond $k^{2 - \epsilon}$. In the special case where all the entries are integers in $O(k)$, this can be improved to $k^{1.5}$ via a recent result by Chi et al. (2022) [1]. In particular, this captures integer weighted graphs without vertex importances. Furthermore, when $k$ is very large, i.e. $k = \Omega(n)$, and the tree-cut sparsifier is balanced, then the overhead is not $n^3$ but just $n^2$. We point out that in theory, we obtain balanced tree cut sparsifiers, and the practical heuristics are typically balanced, too.
>
> [1]: Faster Min-Plus Product for Monotone Instances
> Shucheng Chi, Ran Duan, Tianle Xie, Tianyi Zhang
> STOC 2022

---

> > ### Author Rebuttal · Reviewer_WgKz · 2026-04-01
> >
> > Thank you. My questions have been adequately answered.

---

### Official Review · Reviewer_fYeX · 2026-03-10

**Soundness:** 3
**Presentation:** 3
**Significance:** 3
**Originality:** 3
**Overall Recommendation:** 5
**Confidence:** 4

**Summary:**

This paper deals with the graph label selection problem.
There the input consists of an edge-weighted graph an an integer k, and the goal is to select k vertices L which represent the graph best, that is, one wants to avoid that large cluster of unselected vertices with few connections to the rest of the vertices exist.
This problem is NP-hard.
Thus, the paper provides approximation algorithms for it.
Usually, the goal is to approximate the number of selected vertices k (which was done recently in the literature).
Here, the goal is to still select k vertices, but to approximate the objective function.
For this an algorithm with an approximation guarantee of \tilde{O(log^{1.5}n)} is given.
This is achieved via reduction to tree-cut sparsifiers and maximum flow.
Finally, a proof of concept implementation is provided to show the effectiveness of the approach.

**Compliance With Llm Reviewing Policy:**

Affirmed.

**Final Justification:**

Since I had no questions, I keep my score.

**Key Questions For Authors:**

* abstract say what gets approximated; currently this becomes only clear in the intro
* the abstract should be stand alone, so please move the footnote into the intro (also this sentence seems broken)
* l42 c1 use something as shortcite here
* why |L|\leq k? I think it should never be worse to select k vertices
* l13 c2 w(...) is not defined yet
* l26 c2 please add a formal problem definition (a box etc) and give it the name GLS as it it used later on
* l74 c1 please provide an example for \Psi(L) not being submodular/supermodular
* l55 c2 please separate the refs
* I think the references to the implemented algorithms could already be given in the related work part
* l117 c1 please provide an intuition why this is useful; also A\subseteq C should be a requirement
* l110 c2 please add intuition why this reduction to trees is useful
* lemma 3.6 I am not sure whether this is a new result, doesn't Räcke et al. implicitly do this, or do I miss something here?
* l166 c1 \tau is not defined here: please provide an intuition what \tau will represent later, also provide an intuition for T_{L,\tau}
* l169 c1 here a figure could be added
* l220 c1 this sentence seems broken
* l246 c1 here you assume to have integer weights. Doesn't this imply that in the entire paper you should have this assumption? Or why is is justified to have this assumption only here?
* footnote 4 here the name tree cut sparsifier should be added
* l220 c2 sentence seems broken
* l239 c2 please provide an intuition for the newly added red part
* l263 c2 add that u,v\in V
* lemma 5.6 the case that v only has only one child seems to be missing (although I don't think this is too complicated)
* l353 c1 the name T_{TREECUTSPARSIFIER} should appear later in the paragraph where you discuss the running times
* l371 c1 please cite the heuristics here already
* l375 c1 explain 'bisect'
* algorithm 1 should have a termination condition, i.e., if the size is 1
* l353 c2 it is not clear what 'Fiedler vector' and 'spectral gap' is: at least provide references
* l353 c2 already drop the names of the algorithms here
* it is not clear to me for which values of k you tested the algorithms; for Davies probably every k, but for the larger values it is not clear to me: please provide details in the paper
* l420 c1 please provide an example here in the main part for that running time difference
* l614 c1 important > importance
* appendix B was quite repetitive with the main part since the techniques were identical and only some inequalties used different terms. Maybe one can unify both; or only present the more general result?
* what does the orange/blue areas in Figures 8 and 9 represent?

**Limitations:**

yes

**Strengths And Weaknesses:**

The problem appears to have applications in ML, although it is not used directly.
All proofs appear to be correct to me.
In general, often I missed some intuition on why some steps were done or why certain notation was introduced.
I think more of this would improve the readability a lot (see below for details-for example I likeed the intuition starting in l266 c1, also I liked Figures 2-4 which helped me a lot).
Also, see the comments below for some questions.
I like the idea of the reduction to the tree cut sparsifier, and I also liked the extension to vertex importance.
I also like that the paper provide an implementation, although their data set is quite small (but the number of tested algorithms is large).
Sometimes sentences seem to be broken (see below for soem examples)
Overall, I think the paper could be accepted.

---

> ### Author Rebuttal · Authors · 2026-03-27
>
> We thank the reviewer for their work reviewing our paper, and pointing out various improvements in presentation. We will carefully edit the article.

---

> > ### Author Rebuttal · Reviewer_fYeX · 2026-04-02
> >
> > I had no important questions and there are no important concerns in the other reviews for me.

---

### Decision · Program_Chairs · 2026-04-30

**Decision:**

Accept (regular)

**Comment:**

This paper proposes an approximation approach for the graph label selection problem using tree-cut sparsifiers and dynamic programming. The reviewers appreciated the importance of the problem, the novelty of the approach, and the theoretical contributions. However, they concerned limited comparisons with broader baselines and unclear scalability for massive graphs. In the discussion, the reviewers largely agreed that the paper provides a novel and significant theoretical advance.